# Coaching visits and supportive supervision for primary care facilities to improve malaria service data quality in Ghana: An intervention case study

Amos Asiedu[1,2]*, Rachel A. Haws[3,4]*, Wahjib Mohammed[5], Joseph Boye-Doe[1,2], Charles Agblanya[1,2], Raphael Ntumy[1,2], Keziah Malm[5], Paul Boateng[5], Gladys Tetteh[4,6], Lolade Oseni[4,6]

**1** U.S. President's Malaria Initiative (PMI), PMI Impact Malaria Project, Accra, Ghana, **2** Jhpiego Ghana, Accra, Ghana, **3** Department of International Health, Johns Hopkins Bloomberg School of Public Health, Baltimore, Maryland, United States of America, **4** Jhpiego, Baltimore, Maryland, United States of America, **5** National Malaria Elimination Program, Ghana Health Service, Accra, Ghana, **6** PMI, PMI Impact Malaria Project, Baltimore, Maryland, United States of America

ⓔ These authors contributed equally to this work.
* amos.asiedu@jhpiego.org (AA); rhaws@jhmi.edu (RAH)

## Abstract

Effective decision-making for malaria prevention and control depends on timely, accurate, and appropriately analyzed and interpreted data. Poor quality data reported into national health management information systems (HMIS) prevent managers at the district level from planning effectively for malaria in Ghana. We analyzed reports from a series of 3 data coaching visits conducted at 231 health facilities in six of Ghana's 16 regions between February and November 2021. The visits targeted health workers' knowledge and skills in malaria data recording, HMIS reporting, and how managers visualized and used HMIS data for planning and decision making. A before-after design was used to assess how data coaching visits affected data documentation practices and compliance with standards of practice, quality and completeness of national HMIS data, and use of facility-based malaria indicator wall charts for decision-making at health facilities. The percentage of health workers demonstrating good understanding of standards of practice in documentation, reporting and data use increased from 72 to 83% ($p < 0.05$). By the second coaching visit, reliability of HMIS data entry increased from 29 to 65% ($p < 0.001$); precision increased from 48 to 78% ($p < 0.001$); and timeliness of reporting increased from 67 to 88% ($p < 0.001$). HMIS data showed statistically significant improvement in data completeness (from 62 to 87% ($p < 0.001$)) and decreased error rate (from 37 to 18% ($p < 0.001$)) after completion of the coaching visit series. By the third coaching visit, 98% of facilities had a functional data management system (a 26-percentage-point increase from the second to third coaching visit, $p < 0.0001$), 77% of facilities displayed wall charts, and 63% reported using data for decision-making and local planning. There are few documented examples of data coaching to improve malaria surveillance and service

**Data availability statement:** The datasets used and/or analyzed during the current study are publicly available through Figshare at doi https://doi.org/10.6084/m9.figshare.28765946.

**Funding:** This analysis was supported through the U.S. President's Malaria Initiative (PMI) Impact Malaria project activity (Contract Number 7200AA18C00014) in collaboration with the U.S. Agency for International Development and the U.S. Centers for Disease Control and Prevention. The funders had no role in study design, data collection and analysis, or preparation of the manuscript, but did review and approve the manuscript for submission. The opinions expressed herein are those of the authors and do not necessarily reflect the views of the U.S. President's Malaria Initiative, the U.S. Agency for International Development, the U.S. Centers for Disease Control and Prevention, or other employing organizations or sources of funding.

**Competing interests:** The authors have declared that no competing interests exist.

data quality. Data coaching provides support and mentorship to improve data quality, visualization, and use, modeling how other malaria programs can use HMIS data effectively at the local level.

## Introduction

Efficient and sustainable decision-making for disease prevention and control depends on timely, accurate, well-collected, appropriately analyzed, and effectively interpreted data [1]. In low- and middle-income countries (LMICs), routine data for decision-making are available primarily through each country's national Health Management Information System (HMIS) [2]. It is important that data collected are of sufficient quality to allow regular tracking of progress for monitoring impact, scaling up interventions, and strengthening health systems [3–6]. Routine service data are often poor quality or incomplete, and may be misleading or of limited use for planning and decision-making [3,7]. Concerns about data quality often limit the use of routine service data for prioritizing or targeting program activities, as stakeholders do not trust the integrity of aggregated datasets enough to rely on their analysis for decision-making [7,8]. Recent efforts by ministries of health and stakeholders to improve the quality of HMIS data have prompted interventions at the district level to improve data quality and use, including technological innovations, capacity strengthening, data quality audits, and supportive supervision approaches [9–11].

Gaps in data quality illustrate the need for improved quality of data to inform decision making for malaria prevention, control, and elimination efforts [7]. An assessment of National Malaria Strategic Plans (NMSPs) in 22 countries in sub-Saharan Africa showed that challenges in surveillance, monitoring, and evaluation (SME) included weak health information systems, incomplete data collection, poor data quality, inadequate data management and use and a lack of integration of monitoring systems [12]. Data management issues such as failing to report certain malaria indicators and data reporting delays generally remained unaddressed in subsequent NMSPs. Poor motivation, inadequate supervision, absent feedback mechanisms, and competing disease-specific reporting requests that burden staff contribute to low quality in routine data reporting [13].

Ensuring that high-quality routine data are collected on malaria service provision will be essential to meet aggressive targets set by the World Health Organization (WHO) and RBM Partnership to End Malaria to move countries from malaria control to pre-elimination by 2030 and reduce cases by 90 percent [14,15]. Ghana aims to move six districts from malaria control to pre-elimination by 2025. Studies from countries that have reached the malaria elimination stage have proposed that data-driven decision-making and improved surveillance are essential to address technical, operational, and financial challenges associated with pursuing the goal of elimination, with surveillance as a core intervention [16]. Surveillance is generally weakest in countries with a high malaria burden, underlining the importance of the need for a robust surveillance system [14]. It is essential that quality data on malaria service delivery are available and accessible to policy makers, facility managers, and health workers [17].

In Ghana, routine facility level data are reported monthly in the national electronic HMIS using open-source District Health Information System version 2 (DHIS2) software, which was introduced in 2012 to replace paper-based records and includes data validation, visualization and analysis tools. In Ghana, the DHIS2-based platform is called District Health Information Management System version 2 (DHIMS2). Since DHIMS2 became available, the percentage of health facilities submitting required monthly reports has risen from 11% in 2012 to 91.6% in 2021 [18]. Health workers record provision of malaria-related services using specific recording and reporting tools in DHIMS2. Regular review of DHIMS2 data at the facility level is intended to be used to assess health system performance and for local planning and decision-making, including setting coverage targets, assessing performance on key malaria indicators, and monitoring commodity stock levels to prevent stock-outs.

Supportive supervision visits, health systems performance assessments, and SME Technical Working Group meetings have revealed that health workers in Ghana, especially those posted at primary care facilities, lack understanding of the Ghana Health Service standards of practice regarding health information management and how to record some data elements in registers and reporting forms [19–21]. Additionally, reviews of routine DHIMS2 data have revealed disparities between and within regions in terms of data quality [22]. This problem has led to errors in data capture, extraction, and reporting. As has been documented in other countries, health workers generally underutilize available data visualization tools and techniques to facilitate the use of local facility data for planning and decision making [7,23–25].

Primary health care workers at Community Health Planning and Services (CHPS) compounds and health centers in Ghana play a critical role in collecting, documenting, and reporting malaria service data. Service providers enter patient-level information in facility registers and collate data into monthly reporting forms; they also complete monitoring charts presenting aggregated monthly data to guide clinical meetings and procurement of supplies. Data managers at district level and in some larger and higher-volume facilities conduct data verification and validation of monthly reporting forms and registers with service providers, enter data into DHIMS2, and review data for facility performance and local planning. In facilities without data managers, service providers often perform these roles. Coaching CHPS staff in proper data documentation could improve the quality of the data they collect, reducing the risk of errors, missing information, or duplicate entries [26]. Strengthening capacity in good data management practices can ensure that routine data are reliable and usable for monitoring progress, evaluating programs, and informing best practices. Effective data management can also streamline processes and improve efficiency, freeing up time for health care providers to focus on patient care. Training health workers in basic data visualization techniques can empower them to track progress and make data-driven decisions at the local level. The U.S. President's Malaria Initiative (PMI) Impact Malaria project conceptualized data coaching visits as a means to improve malaria data quality and data use through targeted support to primary facilities with low baseline levels of data quality and completeness.

This case study evaluates how data coaching visits to targeted health facilities affected health workers' knowledge and skills in malaria data recording, extraction and reporting to DHIMS2, and subsequent changes in how managers visualized and used DHIMS2 data for planning and decision making. Findings from this case study illustrate how health workers' skills in malaria data documentation, reporting, and validation can be built and sustained.

## Methods

### Setting and facility selection

Data coaching visits took place in 6 of Ghana's 16 regions where PMI supported malaria elimination efforts. Following the WHO High Burden High Impact (HBHI) strategy, we prioritized primary health facilities—health centers and CHPS—with poor data quality to receive data coaching visits. The intervention prioritized these facilities by using a prioritization matrix to select facilities with data completeness rates below 90 percent on three malaria indicators and error rates above 30 percent on 15 indicators reported to DHIMS2 from August 2020 to January 2021. Completeness rate was calculated as the

number of data elements reported in DHIMS2 divided by the total number of expected routinely collected data elements. The three sentinel indicators on which data completeness was assessed were: 1) number of suspected malaria cases at outpatient department (OPD) tested for malaria using rapid diagnostic tests (RDT) or microscopy; 2) number of confirmed malaria cases, disaggregated by test type: RDT and microscopy; and 3) confirmed uncomplicated malaria cases treated with antimalarial artemisinin-combination therapy (ACT). An error was defined as an discrepancy of +/-10 percent between the OPD register and the national DHIMS2 on any of 15 validated malaria indicators (S1 Table). Error rate was calculated as the number of indicators flagged with any error during the data validation exercise divided by the total number of validated indicators. A total of 231 out of 605 total facilities in 52 districts across 6 regions were selected for the intervention using the prioritization matrix.

**Intervention implementation**

Prior to the coaching visit series, PMI Impact Malaria staff, in collaboration with National Malaria Elimination Program and Policy, Planning, Monitoring & Evaluation Division (PPMED) staff from the Ghana Ministry of Health, trained regional and district level malaria focal persons and health information officers on the data coaching tools and DHIMS2 standards of practice.

Data coaching visits to facilities were designed to strengthen data managers' and service providers' competency in proper documentation in registers, accurate transfer of data onto reporting forms and into DHIMS2, and ability to use tools that track and present data on malaria indicators. Using the Plan-Do-Study-Act (PDSA) continuous improvement cycle [27], an iterative quality improvement approach that identifies problems and develops and tests potential solutions—"change ideas"—to these problems, managers were guided during these visits to develop change ideas to address identified data gaps in their facilities and districts (Fig 1).

Two teams made up of three staff (the district and regional health information officer and a malaria focal person) provided data coaching visits to all prioritized facilities, with each team visiting two facilities per day and spending an average of 4 hours in each facility. During each coaching visit, health workers were given a pretest on knowledge about the various malaria-related registers and reporting forms, variables in registers (e.g., definition of old versus new cases), tools for data validation, and use of facility-based wall charts for planning. First, coaching teams reviewed facility data with health workers to compare the source (register) to the data reported to DHIMS2. Next, they coached health workers on register variables, completion of reporting forms, and data validation following DHIMS2 standards of practice. Coaching teams then presented a facility-based wall chart to explain proper documentation, reporting and the use of wall charts for local planning, demonstrating for participants how to complete and use malaria indicator wall charts to track the performance of malaria indicators at the facility level. During these visits, coaching teams also guided managers to develop change ideas using the PDSA continuous improvement cycle to address identified data gaps in their districts and facilities. Two additional coaching visits were conducted three and nine months later using the same assessment tools to assess improvement in data reporting, facility display of updated wall charts, and reported use of the dashboard and wall chart for decision-making.

**Evaluation design**

To evaluate the intervention, we used a before-after design including four components: facility performance monitoring and data quality assessment, DHIMS2 data quality review, observation of evidence of data display and use at the facility level at each coaching visit, and a satisfaction survey of those who participated in the data coaching activities. The facility performance monitoring and data quality assessment monitored facility-based data management systems and compliance with standards of practice for DHIMS2 data entry. Longitudinal analysis of coaching visits over time and analysis of key malaria indicators in DHIMS2 were used to assess longer-term changes in data documentation, reporting, and use for decision-making. We also evaluated health workers' use of data for local planning over the course of the coaching

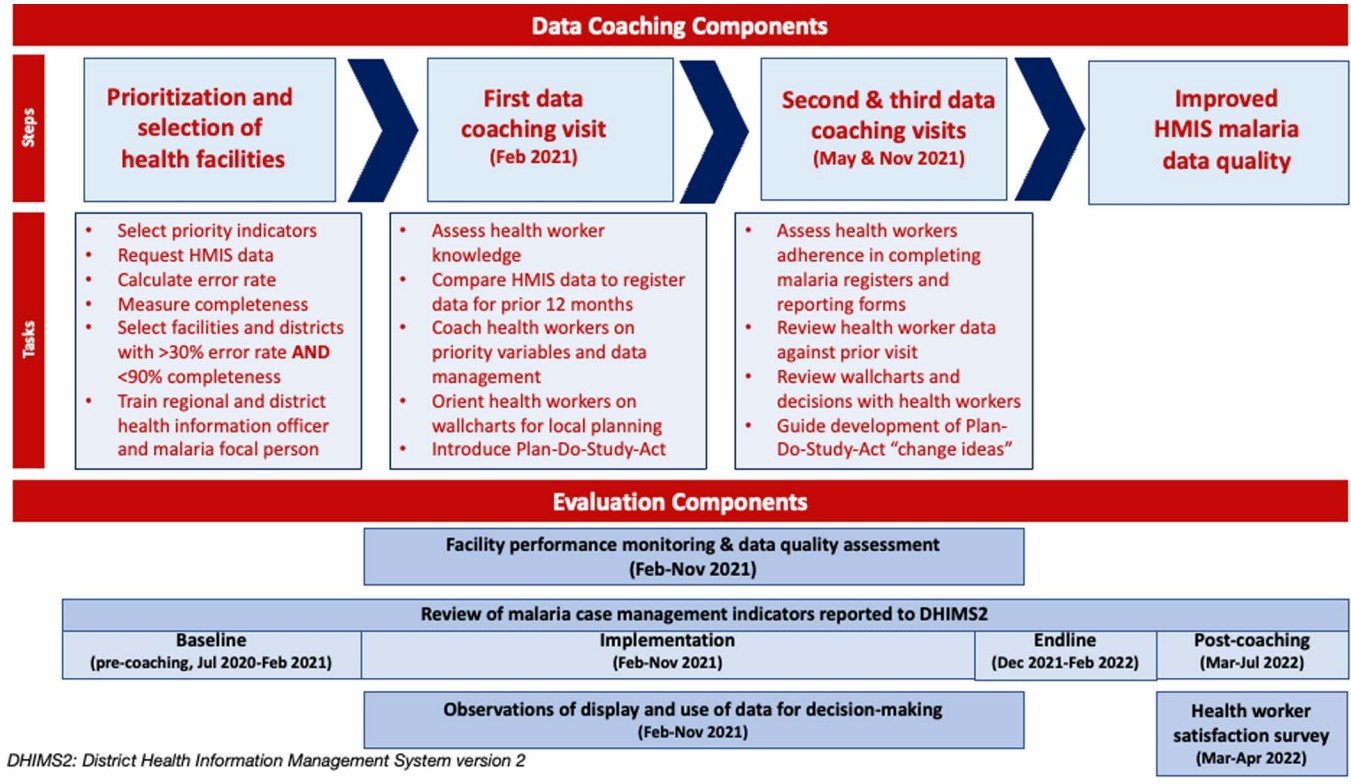

**Fig 1. Implementation framework.**

visit series, and the satisfaction survey was used to to evaluate the experiences and perceptions of health workers and regional and district level supervisors who had participated in the data coaching activities.

**Facility performance monitoring and data quality assessment.** During each of the three coaching visits (February 2021, May 2021, and November 2021), we assessed whether facilities had functional data management systems conforming to GHS standards of practice, as well as health worker understanding of the GHS standard performance monitoring system and data quality. The assessment evaluated health worker understanding of the facility data management system and compliance with GHS standards of practice for documenting data in DHIMS2. Regional and district level health information officers conducted the assessments using the standardized USAID Data Quality Assessment (DQA) checklist tool [28] programmed on the KoboCollect App [29] and stored all collected data in the PMI Impact Malaria database. The DQA assessed data quality across five dimensions as outlined by USAID: validity (data measures what it is intended to measure), reliability (consistent over time and varying conditions), integrity (free of bias, manipulation, and error), precision (detailed enough to make informed management decisions), and timeliness (collected regularly so available and current when needed for decision-making). To measure improvement in the performance monitoring system of the facilities, we compared DQA scores from each coaching visit to the prior coaching visit.

**DHIMS2 data quality review.** We also assessed how improvements in the facility performance monitoring system translated to changes in data reporting completeness and error rates in DHIMS2 data after the coaching visit series by comparing seven routinely collected indicators from a period before coaching visits (0–6 months prior to the first coaching visit) to data from monthly reporting forms and registers from each facility averaged for the period during the coaching visit series (February-November 2021), immediately after the intervention (December 2021-February 2022) and several months after the coaching visit series had ended (March-June 2022). These seven indicators together represent

a malaria case management cascade at the facility level: suspected cases of uncomplicated malaria, suspected cases of uncomplicated malaria tested, confirmed malaria cases, outpatients treated with antimalarials, outpatients treated with artemisinin-combination treatments, suspected uncomplicated malaria cases tested using RDTs, and suspected uncomplicated malaria cases tested using microscopy. These routinely collected quantitative facility-level service data were reported monthly by all 231 facilities to DHIMS2.

**Observation of data display and use.** At every coaching visit, we assessed whether and how facilities visualized data for decision making and data use for planning at every coaching visit. We used the KoboCollect App to collect observations on whether wall charts were displayed, updated, and being used for decision-making about procurement of supplies, workflow, adherence to guidelines, in-service trainings, and health education priorities. All data were stored in the PMI Impact Malaria database.

**Satisfaction survey.** We conducted an after-action review from March-April 2022 using an online satisfaction survey. We sent a Google Forms link via WhatsApp to health workers at facilities that had received coaching visits (N = 462), as well as health information officers (N = 52) and malaria focal persons (N = 52). The survey link was closed in March 2022 after a convenience sample of 250 participants had responded anonymously to the survey. Facilities were randomly selected, but inclusion criteria included smartphone ownership and internet access, which limited the sample by approximately 40%. The survey assessed the opinions, experience, and perspectives of health workers, health information officers, and malaria focal persons about how data coaching affected facilities' performance systems, data quality, use of data for decision making and data reporting in DHIMS2.

## Data analysis

**Facility performance monitoring and data quality assessment.** For each component/indicator, a composite performance score across facilities was computed using a simple arithmetic sum, adding all questions with a "correct/yes" response, dividing by number of facilities, and converting to a percentage to compute a mean score across facilities. We used a two-sample t-test at a 5% level of significance to detect differences in mean assessment scores on each component between first and second coaching visits, as well as between the second and third coaching visits.

**DHIMS2 data quality review.** We used a paired t-test using Stata/IC version 14.2 to identify statistically significant changes over time in routine monthly DHIMS2 indicators relevant to malaria case management as a proxy for quality [30]. The analysis measured changes in data reporting completeness and error rates using the period 0–6 months prior to the first coaching visit as a baseline, comparing this to the coaching visit implementation period (March-November 2021), the three-month period immediately following the coaching visit series as a proxy endline (Dec 2021-Feb 2022), and several months after the coaching visit series had ended (March-Jun 2022) to assess whether improvements were sustained. Each period was compared statistically to the period immediately prior.

**Observation of data display and use.** We measured improvement in the accurate updating of facility-based wall chart and data use for local planning (e.g., ITN distribution, reducing delays in service provision, and identifying staff training needs) by comparing each coaching visit assessment score to the score from the prior coaching visit.

**Satisfaction survey.** The study team read through survey responses and highlighted emergent themes individually in Atlas.ti Version 7.5 using a thematic analysis approach [31]. Themes and codes were reviewed and discussed throughout the analytic process, and refined or adapted as needed based on emergent information from the survey responses. Data were analyzed by type of participant, then discussed and summarized, noting differences between types of participants. The team then collated and reached consensus on themes through team discussions.

## Data management and confidentiality

No personally identifiable information was collected during the assessment. All secondary data included in the study were routine malaria service data that are available to authorized users or on request from the Ghana Health Service. Project

monitoring and evaluation data gathered during the coaching visits included providers' sex, cadre, data coaching performance assessment score, and evidence of data use for decision making. Names of providers and their facilities were not collected. The qualitative satisfaction survey was anonymous: names and facility identification of health workers were not collected as part of the survey. All data collected and retrieved for the study were stored securely in the password-protected PMI Impact Malaria database.

## Ethical considerations

This assessment obtained a non-human subjects research determination from the Johns Hopkins Bloomberg School of Public Health Institutional Review Board, Maryland, USA (IRB# 21543). Study participants who completed the satisfaction survey gave their written informed consent.

## Results

### Participating facilities, health workers, and data coaches

A total of 14 regional-level and 104 district-level supervisors, including malaria focal persons and health information officers from 52 districts across six regions, served as coaches and visited facilities. All 231 targeted health facilities received a coaching visit; 833 health workers (327 male; 506 female) participated in coaching visits at these facilities. The second and third coaching visits each reached 1211 health workers (578 male; 633 female). The same number of health workers were reached at each coaching visit, though specific individuals reached at these later coaching visits varied because of staff turnover and availability on the day of the visit. Twenty-three percent (N = 54) of the 231 targeted facilities were health centers, and the rest (N = 177) were CHPS.

### Performance monitoring and data quality assessment

Almost all performance monitoring scores and data quality scores improved from the first to the second coaching visit (Table 1). The percentage of health workers demonstrating good understanding of documenting, reporting and use increased from 72 to 83% (p < 0.05). Sixty percent of facilities were using standard malaria registers at the first coaching visit, but by the third visit, 95% of the facilities were using standard malaria registers. While the percentage of facilities with functional data storage and management systems only increased slightly between the first and second coaching visit, 98% of facilities had a functional data storage and management system by the third coaching visit, up from 72% at the second coaching visit (p < 0.0001).

Compliance of health workers with almost all standards of practice for DHIMS2 reporting had improved at the second coaching visit (Table 1). Improvements were statistically significant for four discrete skills: ability to differentiate between a new and old case of malaria, reporting only new cases on the malaria section of OPD morbidity, reporting only old cases under "repeat attendances" on OPD morbidity returns, and reporting case management cascades (number suspected, tested with RDT or microscopy, confirmed positive, treated with ACT) in OPD morbidity returns. Data quality also improved by the second coaching visit, including increased reporting of errors and correcting mistakes, data reviewed for accuracy, procedures used to ensure accurate data from registers to reporting, and avoiding double counting of malaria data.

### DHIMS2 data quality and completeness

Compared to the first coaching visit, the quality of data that facilities reported to DHIMS2 at the second coaching visit improved significantly across three of the five DQA dimensions of quality (Table 2). Reliability increased from 29% to 65% (p < 0.001); precision increased from 48% to 78% (p < 0.001); and timeliness of reporting increased from 67% to 88% (p < 0.001). By the third coaching visit, reliability had increased from 65% to 89% (p < 0.05). Integrity also improved from 66% at the second coaching visit to 90% at the third coaching visit (p < 0.05).

**Table 1. Changes in performance, adherence to standards of practice, and data quality after data coaching visits.**

| Performance Monitoring System and Data Quality Assessment | First coaching visit mean score (N = 833, %) | Second coaching visit mean score (N = 1211, %) | Mean difference (Second vs. first coaching visit) (percentage points, 95% CI) | p-value | Third coaching visit mean score (N = 1211, %) | Mean difference (Third vs second coaching visit) (percentage points, 95% CI) | p-value |
|---|---|---|---|---|---|---|---|
| **Monitoring System** | | | | | | | |
| Staff demonstrate good understanding of data recording, reporting and use | 72% | 83% | **11 (2, 2)** | 0.046 | 89% | 6 (−6, 18) | 0.353 |
| Functional facility data storage and management system | 68% | 72% | 4 (−7, 15) | 0.501 | 98% | **26 (16, 36)** | **<0.001*** |
| **DHIMS2 Standards of Practice Compliance** | | | | | | | |
| Availability of Ghana Health Service DHIMS2 Standards of Practice | 36% | 48% | 12 (−0.4, 24) | 0.057 | 50% | 2 (−16, 20) | 0.825 |
| Staff able to differentiate between a new and old case* of malaria | 61% | 89% | **28 (18, 37)** | **<0.001*** | 96% | 7 (−2, 15) | 0.169 |
| Staff report only 'New Cases' of malaria for the malaria section of outpatient department monthly reporting form | 83% | 94% | **11 (4, 18)** | **0.012*** | 89% | −5 (−15, 5) | 0.301 |
| Staff report only 'Old Cases'* of malaria under repeat attendances in the outpatient department monthly reporting form | 73% | 88% | **15 (6, 24)** | **0.005*** | 93% | 5 (−5, 15) | 0.364 |
| Reporting case management cascade (number suspected, tested with rapid diagnostic test or microscopy, confirmed positive, treated with artemisinin-based combination therapy) in outpatient department monthly reporting form, with first three indicators disaggregated by pregnancy status | 80% | 93% | **13 (5, 20)** | **0.005*** | 93% | 0 (−9, 9) | 1.000 |
| Errors in malaria data reported, have been tracked to their original source and mistakes corrected | 89% | 83% | 6 (−15, 3) | 0.166 | 89% | 6 (−6, 18) | 0.178 |
| Data are reviewed for accuracy before sending to the next level/entry into DHIMS2 | 42% | 65% | **23 (11, 35)** | **<0.001*** | 82% | **17 (2, 32)** | **0.039*** |
| Appropriate procedures used to ensure accurate data transferred from registers to reporting forms | 53% | 66% | **13 (9, 25)** | **0.041*** | 89% | **23 (10, 36)** | **0.004*** |
| Appropriate procedures for avoiding double counting of malaria data | 45% | 65% | **20 (10, 32)** | **0.002*** | 80% | 15 (−1, 30) | 0.071 |
| Systems and procedures in place to ensure timely reporting of malaria data | 73% | 83% | 10 (−1, 20) | 0.068 | 89% | 6 (−6, 18) | 0.354 |

* An "old case" is defined as a client who tested positive for malaria within 28 days of the first episode.

Data in DHIMS2 for the study facilities showed statistically significant improvement in data completeness (increasing from 62% before coaching visits to 87% during coaching visit implementation (p < 0.001)) and decreased error rate (falling from 37% before coaching visits to 18% during coaching visit implementation (p < 0.001)) (Table 3). Error rates continued to drop after coaching visits ended compared to the period of implementation (p < 0.05). Completeness also improved, but the difference was not statistically significant.

**Table 2. Changes in USAID data quality assurance checklist scores on data quality dimensions after data coaching.**

| Data Quality Assessment Dimension | First coaching visit score (N=833) | Second coaching visit* score (N=1211) | Percentage-point difference (second vs. first coaching visit) | p-value | Third coaching visit* score (N=1211) | Percentage-point difference (third vs. second coaching visit) | p-value |
|---|---|---|---|---|---|---|---|
| | | | Mean (95% CI) | | | Mean (95% CI) | |
| Validity | 87% | 94% | +7 (−8, +14) | 0.079 | 94% | 0 (−11, +11) | 1.000 |
| Reliability | 29% | 65% | **+36 (+24, +48)** | **<0.000*** | 89% | **+24 (+11, +37)** | **0.003*** |
| Integrity | 54% | 66% | +12 (−2, +24) | 0.059 | 90% | **+24 (+11, +37)** | **0.002*** |
| Precision | 48% | 78% | **+30% (+19, +41)** | **<0.000*** | 89% | +11 (−2, +23) | 0.116 |
| Timeliness | 67% | 88% | **+ 21% (+11, +31)** | **<0.000*** | 91% | +3 (−8, +14) | 0.597 |

Second and third coaching visits were 3 and 9 months after first visit, respectively. CI: confidence interval.

## Data use for decision making and local planning

At the first coaching visit, no health facilities had a facility wall chart or any other means to visualize and use their data for decision making and local planning. However, by the second coaching visit, 75% of facilities displayed wall charts; this figure rose to 77% at the third coaching visit (Fig 2). At the second coaching visit, 84% of new health workers had been oriented to the use of wall charts; this proportion fell slightly to 80% at the third coaching visit. By the second coaching visit, 60% of facilities reported using data for decision making and local planning; this rose further to 63% at the third coaching visit. None of the differences between the second and third coaching visits were statistically significant.

## Health worker satisfaction

A total of 250 health workers, malaria focal persons, and health information officers responded to the qualitative survey to share their experiences with the coaching visits (Table 4). Most were male (73.5%). More than half (63%) of the respondents were aged 30–39 years, with 31% aged 20–29 years and 6% aged 40–49 years. More than half (53%) had between

**Table 3. DHIMS2 data quality over time.**

| DHIMS2 Data Reporting | Before coaching (0–6 mos pre-coaching) n=138 600 | Implementation (9-mo period during coaching visits) n=140 122 | Percentage-point difference during vs. before coaching | Endline (1–3 months after last coaching visit) n=139 433 | Percentage-point difference endline vs during coaching | Post-coaching (4–7 mos after last coaching visit) n=139 433 | Percentage-point difference post-coaching vs endline |
|---|---|---|---|---|---|---|---|
| | Mean % (+/- SD) | Mean % (+/- SD) | Mean % (95% CI) p-value | Mean % (+/-SD) | Mean % (95% CI) p-value | Mean % (+/-SD) | Mean % (95% CI) p-value |
| Completeness | 62 (1.4) | 87 (0.9) | **+ 25 (+8, +15) p<0.001** | 94 (1.0) | + 7 (−5, 13) p=0.234 | 95 (1.8) | +1 (−3, 8) p=0.324 |
| Error rate | 37 (1.7) | 18 (1.4) | **-17 (-25, -13) p<0.001** | 10 (0.7) | **−8 (−18, −7) p=0.035** | 8 (1.0) | −2 (−6, 4) p=0.180 |

DHIMS2: District Health Information Management System version 2, SD: standard deviation; CI: confidence interval

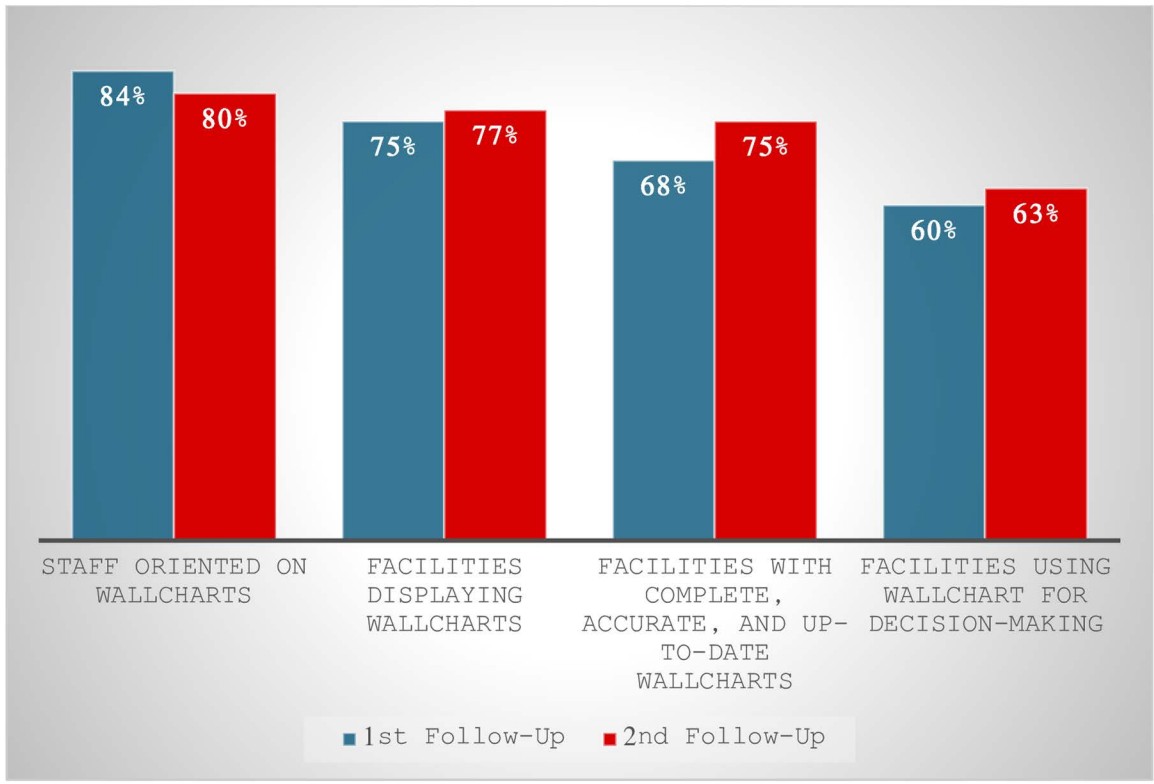

**Fig 2. Wall chart availability and use after data coaching (N = 231 facilities, N = 1211 staff).**

**Table 4. Background characteristics of participants (N = 250).**

| Variable | Response(s) | N (%) |
|---|---|---|
| Gender | Female | 66 (26.5) |
| | Male | 184 (73.5) |
| Age (year(s)) | 20-29 | 78 (31.0) |
| | 30-39 | 156 (63.0) |
| | 40-49 | 16 (6.0) |
| | 50-59 | 0 (0) |
| Years of work experience | <1 | 66 (26.5) |
| | 1-5 | 133 (53.0) |
| | 6-10 | 51 (20.5) |

1 and 5 years of work experience. Their responses reveal how coaching visits improved data documentation and use by health workers at the CHPS, health centers, district and regional levels.

**Improved data quality.** Coaching participants roundly agreed that the coaching visits had helped to improve the quality of data documented and reported at their facility. They reported that the coaching had improved their understanding of documentation in the registers, reports, and the data collection process in the facility. Some added that the coaching had helped them to validate their data, and had improved the timeliness of their reporting. Service providers indicated that the wall charts helped them to identify areas for improvement and that being able to see the number of malaria cases suspected, tested and given treatment helped them follow treatment protocols.

*"The training has helped to improve documentation in the consulting room register and led to timely submission of data."* – Health information officer

*"In fact, data coaching visits have increased our knowledge on how to record data properly in the consulting room register, antenatal care register, outpatient register, and reports compilation in our facility."* – Enrolled Nurse

**Adherence to malaria prevention and treatment guidelines.** Respondents reported that wall charts had significantly improved data accuracy, malaria diagnostic accuracy, and staff awareness through promoting improved adherence to guidelines:

*"Since the implementation of wallcharts to improve adherence to malaria prevention and treatment guidelines, we have observed several improvements and changes among our staff. Here are some notable improvements: Adherence to Test, Treat, Track (T3), reduction of presumptive treatment, improved IPTp and ITN provision, and improved referral practices... Overall, the implementation of wall charts has positively impacted staff adherence to malaria prevention and treatment guidelines. It has promoted evidence-based practices, reduced presumptive treatment, increased adherence to IPTp and ITN provision, and enhanced appropriate referral practices."* – Senior Technical Officer

*"Of course, the wall charts have helped facilities adhere to the 3Ts in malaria case management. Facilities can determine the number of malaria cases tested and treated at a glance. And this guides them to monitor their RDT stock levels and put in requisition as soon as possible."* – Health Information Officer

*"Since the use of the wall chart, facilities learn to probe further into conditions reported to them and with the appropriate tests conducted. All positive malaria cases are put on antimalarials and asked to come back for review."* – Nurse

**Local planning.** Some service providers reported using wall charts to monitor trends in indicators, both for planning and forecasting. Respondents noted that monitoring trends in indicators gave them advance notice of areas that needed improvement, and that monthly monitoring improves district indicators and identifies errors. Several respondents mentioned that action plans now consider positivity rates and the number of antenatal care registrants in their facilities.

*"The wall charts help us to identify indicators where we are not meeting the target and to plan…it helps us to monitor trends of health indicators."* – Health Information Officer

*"The skills and knowledge I have acquired help me to prepare an action plan for the facility and also help in providing comprehensive care."* – Registered Nurse

*"We have improved our district's indicators due to monitoring on a monthly basis, also enabling us to identify errors with our data before the year ends."* – Health Information Officer

Respondents frequently mentioned using wall charts to monitor key commodities such as ITNs and antimalarials. Respondents explained that the use of IPTp for guiding commodity allocation has reduced delays in commodity requests and helped monitor commodity stock levels, enabling timely restocking and demand forecasting. Health Information Officers mentioned that learning to monitor stock levels and trends on a monthly rather than annual basis has improved data quality and forecasting for commodities like ITNs and SP, preventing shortages. Wall charts also help facilities notice when commodities are running out, allowing for restocking to avert stockouts:

*"The coaching has helped us to enter data easily and it's making our work very simple using the wall chart. At a glance at the graph, I can track our performances and coverage. It makes it easier to do my division and subtraction on the stock using the graph. We are now able to do our requisitions on time so that there will be no shortage of SP for IPTp when pregnant women come for ANC."* – Registered Nurse

*"Plans for procurement at the facility now take into consideration the positivity rate recorded whereas ITNs factors in the Number ANC Registrants at a specified period."* – Health Information Officer

*"Using the wall chart has been helpful to the facility in monitoring their stock level of commodities … they do not run out of commodities."* – Health Information Officer

## Discussion

Coaching, supervision, and mentorship approaches have been used to improve health data reporting, analysis, and use in many health areas, but have been rarely detailed in the malaria literature, particularly at the facility level. This study, along with an emerging body of programmatic research, demonstrate that data coaching and other sustained data-focused supportive supervision and mentorship efforts can help improve data literacy, data quality, data visualization, data-driven decision making, and data culture.

The only other documented examples of data coaching for malaria programs come from Madagascar and Uganda. In Madagascar, PMI Measure Malaria developed a data coaching guide that defined the role of a coach, conduct of coaching visits, and performance indicators, and also provided a template to document action items [32]. Coaching visits were associated with improved staff skills and confidence using DHIS2, improved data quality (completeness rose from 86 to 91 percent; timeliness rose from 54 to 81 percent), analysis, and visualization. In Kayunga, Uganda, a pilot of an intervention that included in-service training in 5 health centers, mentored continuous improvement with PDSA cycles, learning sessions, and coaching reported improvements in data completeness [33]. Health workers appreciated coaching, but some expressed frustration with perceived additional workload from data collection and continuous improvement activities. Financial incentives for participation helped offset this frustration.

Interventions to improve the quality of DHIMS2 data can take technical approaches (e.g., improving paper data collection forms, developing electronic tools, mHealth approaches employing tablets or smartphones, data quality audits/assessments, data storage) or behavioral approaches (e.g., short- or long-term training, task-shifting, supportive supervision, stakeholder engagement, incentives, standardized protocols) [11]. Of all interventions, training is the most commonly implemented component, and is widely shown to have a positive impact on data quality. A study of primary health facilities in Nigeria associated training on data management with statistically significant increases in completeness of reporting, accuracy, timeliness, and feedback [34]. However, a systematic review showed that interventions with multiple components that include training showed greater benefit than those that tested single intervention components [11]. A cross-sectional study of health centers in Chad found that data errors were associated with high workload, stockouts, and unavailability of required data collection tools, as well as the absence of a health technician and dedicated staff for data management [35]. These visits also offer opportunities to utilize electronic tools that increase data completeness and reduce errors [21]. Across Tanzania, electronic Malaria Services and Data Quality Improvement tools are used during supportive supervision visits to assess the quality of routinely collected malaria data reported from health facilities; the tools include a data quality assessment component, and facilities develop action plans to address deficiencies in service quality identified by the tools [36]. Our study showed that data coaching fostered the implementation of functional data management systems in accordance with standards of practice that may strengthen data collection at all levels. Data coaching can also offer an opportunity to evaluate working conditions for clinic personnel, as improvements in workload, staffing, and facility readiness may also improve DHIMS2 data quality.

Strategies that visualize and track health facility indicators—including wall charts, scorecards, and dashboards—are increasingly popular tools to encourage interpretation and use of data for improved performance at the facility and district level. In our study, data coaching activities promoted the creation and display of wall charts. While studies across health sectors generally demonstrate that data visualization approaches can improve the quality and completeness of data, and improved access to and use of data at district, region, and national levels [25,37,38], their effectiveness in promoting

effective use of data at the facility level remains largely unmeasured [39]. Most examples are of data being reported upward in the health system, and how data are used is rarely detailed. In Nigeria, a national project to promote use of dashboards using routine data for decision making led to increased completeness of DHIS2 reports from 53% to 81% over three years of the project, but primary healthcare governance structures and limited human resource capacity constrained use of data for decision making [40]. In Zanzibar, Jhpiego worked with the Ministry of Health to construct a dashboard integrating multiple sources of malaria-related data [41]. The dashboard identified decision-making errors addressed through supportive supervision visits, as well as underutilization of primaquine at private health facilities, prompting the Ministry of Health to develop job aids and antimalarial record books. In Malawi, Hazel et al documented improved reporting consistency and use of data-display wall charts for integrated community case management after a data quality and use package was introduced that included a wall chart template for data display, training for health surveillance assistants and supervisors, and a participatory approach that engaged community and facility health workers in program improvement [25]. Some participants reported using data in wall charts for program improvement—for example, to track stockouts. Sustained support and supervision appear crucial for the success of this modality, as use of the tools declined once scaled up nationally from a smaller pilot. Similarly, use of laminated or electronic wall charts developed in 20 countries under the Maternal and Child Survival Program was sustained in only half of these countries after the project concluded [42]. By documenting adoption of wall charts and reports of data use by managers, our study contributes evidence that data coaching visits can empower managers and providers at the facility level, but leadership should consider sustainability of these efforts.

More importantly, surprisingly little research has investigated what data are actually used by managers of primary health care facilities for decisions about service delivery [43]. Although the DHIS2 software platform is used to support HMIS in more than 70 countries, including Ghana's DHIMS2, underutilization or non-use of data by facilities is common [6]. Few examples of DHIS2 data use at the local level have been captured in the academic or programmatic literature, particularly for malaria [44]. While data can shape effective policy and evidence-based practices, neither possession of nor access to high quality data guarantees its use for decision-making, as a host of organizational factors shape data use [45]. For example, in Tanzania, Mboera et al found that while two-thirds of facility respondents used HMIS data, only 38.5% routinely analyzed it due to poor data utilization and inadequate data analysis in most districts and health facilities, stemming from insufficient resources, inadequate incentives and supervision, and nonexistent standard operating procedures for data management [46]. A mixed-methods study of prevention of mother-to-child transmission of HIV programs in South Africa found that while 53% of study facilities used data, "use" primarily meant reporting program outputs to the provincial level, not informing decisions and planning [47]. The study found an absence of a culture of data use, low trust in the data, and poor data literacy among program and facility managers.

By promoting data culture, which involves designing systems that support work practices and provide forums for conversations around data, data coaching can change health worker perceptions [44]. Our findings suggest that data coaching is supporting a culture of data use in Ghana to encourage effective commodity management at facility and district levels, as well as improved case management at the facility level. In Malawi, Kumwenda et al found that health workers providing HIV services saw data management as part of their role, but generally viewed these responsibilities as secondary to provision of clinical care [48]. Strategies that motivate health workers to adopt data quality improvement practices at health facilities can promote a data culture that in turn leads to effective data use at the facility level; where workloads are heavy or where private sector providers are involved, financial incentives or recognition can nurture a culture of data use [33,49]. Even where health workers articulate the need for an effective health data management system, they may lack technical capacity, policy support, infrastructure, or facility readiness, particularly in rural areas [50]. Regular data review meetings, supportive supervision, provision of feedback, and training on computer skills and data collection tools are strategies that can help health workers comprehend the significance and utility of data they collect [37,48].

Sustainable investment in improving data quality requires a systems approach and appropriate governance at all levels of the health system. To make locally appropriate and timely decisions about services, workforce, and procurement, facilities need disaggregated data and managers empowered to create enabling environments [43]. Improving the quality of health data requires continuous effort from health facilities, feedback mechanisms between programs, health officers, and providers, and a willingness to make course corrections at all levels [6]. Going forward, the impact of data coaching approaches to improve malaria services may be enhanced and made more sustainable through the use of supportive technologies. Electronic tools to improve data quality and data use are increasingly used to supplement training efforts. While many low- and middle-income countries have moved in recent years toward electronic data collection systems to streamline data collection [35], these systems do not ensure overall data quality, nor do they directly promote effective understanding and use of data, particularly at the local level [51]. A review of data quality and data use in routine HMIS found that interventions to improve data quality were most effective when they combined capacity building activities such as training, coaching, or supportive supervision with technology enhancement, and that data use for planning rose when interventions promoting data availability were combined with technology enhancement [3]. Coupling data coaching with electronic tools for data collection, analysis, and visualization may achieve even greater gains in data quality and data use.

Activities were carried out on a limited budget as part of program activities. While we did not assess quality of individual consultations, measuring provider skills and knowledge in conjunction with case management cascade indicators can serve as a proxy for quality of case management. We did not evaluate or investigate the organizational culture, infrastructure, or staffing at each study facility, which Daneskohan et al note may have a significant influence on how health workers collect, report, and use data [52]. Changes in HMIS error rates and data completeness may be attributable to programmatic or ecological factors other than the data coaching intervention. We relied on self-reports from providers and managers about how data were being used, which may be inflated due to social desirability bias. Additionally, the intervention did not include enhanced technologies that have been shown to magnify the impact of data quality improvement initiatives; this is an area ripe for future research.

## Conclusions

In primary health care facilities in Ghana, a data coaching intervention improved data documentation, management, and reported use for local planning. Participatory development of facility-specific action plans using the PDSA continuous improvement approach promoted local investment in quality improvement. While there are few documented examples of data coaching to improve the quality of malaria surveillance and service data in the academic literature, data coaching—or similar efforts to provide targeted supportive supervision and mentorship on data quality, visualization, and use—offers a model for other malaria intervention programs to document and use HMIS data effectively to improve service quality at the facility level.

## Supporting information

**S1 Table. Indicators and definitions for error calculations.**
(DOCX)

## Acknowledgments

The authors are grateful for the support and expertise of the PMI Impact Malaria in Ghana team. We would especially like to thank the Ghana National Malaria Elimination Program, as well as regional, district, and facility health management teams for their partnership, support, and leadership. The authors would also like to thank Dr. Mark Kabue for his guidance in conceptualizing the manuscript.

## Author contributions

**Conceptualization:** Amos Asiedu, Rachel A. Haws, Wahjib Mohammed, Raphael Ntumy, Paul Boateng, Gladys Tetteh, Lolade Oseni.

**Data curation:** Amos Asiedu, Joseph Boye-Doe.

**Formal analysis:** Amos Asiedu, Rachel A. Haws, Joseph Boye-Doe, Lolade Oseni.

**Funding acquisition:** Gladys Tetteh, Lolade Oseni.

**Investigation:** Amos Asiedu, Charles Agblanya.

**Methodology:** Amos Asiedu, Rachel A. Haws, Lolade Oseni.

**Project administration:** Amos Asiedu, Charles Agblanya, Raphael Ntumy, Gladys Tetteh, Lolade Oseni.

**Software:** Joseph Boye-Doe.

**Supervision:** Amos Asiedu, Rachel A. Haws, Wahjib Mohammed, Raphael Ntumy, Keziah Malm, Gladys Tetteh, Lolade Oseni.

**Visualization:** Amos Asiedu, Rachel A. Haws, Lolade Oseni.

**Writing – original draft:** Amos Asiedu, Rachel A. Haws, Lolade Oseni.

**Writing – review & editing:** Amos Asiedu, Rachel A. Haws, Wahjib Mohammed, Joseph Boye-Doe, Charles Agblanya, Raphael Ntumy, Keziah Malm, Paul Boateng, Gladys Tetteh, Lolade Oseni.

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
