## [Decision Letter · Decision Letter 0]

16 Feb 2025

PGPH-D-24-01778

Coaching visits and supportive supervision for primary care facilities to improve malaria service data quality in Ghana: an intervention case study

Dear Dr. Haws,

Thank you for submitting your manuscript to PLOS Global Public Health. After careful consideration, we feel that it has merit but does not fully meet PLOS Global Public Health’s publication criteria as it currently stands. Therefore, we invite you to submit a revised version of the manuscript that addresses the points raised during the review process.

Please note that we have only been able to secure a single reviewer to assess your manuscript. We are issuing a decision on your manuscript at this point to prevent further delays in the evaluation of your manuscript. Please be aware that the editor who handles your revised manuscript might find it necessary to invite additional reviewers to assess this work once the revised manuscript is submitted. However, we will aim to proceed on the basis of this single review if possible. 

Could you please revise the manuscript to carefully address the concerns raised?

We look forward to receiving your revised manuscript.

Kind regards,

Helen Howard

Staff Editor

Journal Requirements:

1. In the online submission form, you indicated that "The datasets used and/or analyzed during the current study are available from the corresponding author on reasonable request.". 

a. In a public repository, 

b. Within the manuscript itself, or 

c. Uploaded as supplementary information.

Additional Editor Comments (if provided):

Reviewers' comments:

Reviewer's Responses to Questions

**Comments to the Author**

1. Does this manuscript meet PLOS Global Public Health’s publication criteria ? Is the manuscript technically sound, and do the data support the conclusions? The manuscript must describe methodologically and ethically rigorous research with conclusions that are appropriately drawn based on the data presented.

Reviewer #1: Partly

2. Has the statistical analysis been performed appropriately and rigorously?

Reviewer #1: Yes

3. Have the authors made all data underlying the findings in their manuscript fully available (please refer to the Data Availability Statement at the start of the manuscript PDF file)?

Reviewer #1: No

4. Is the manuscript presented in an intelligible fashion and written in standard English?

Reviewer #1: Yes

5. Review Comments to the Author

Reviewer #1: General comments

- A variety of terms are used to describe the various stages of the study and data collection: “baseline”, “post-intervention”, “first follow-up”, “second follow-up”, “midline”, “endline”, “post-endline”. To improve clarity, I strongly recommend using consistent terms to describe each period when outcomes are assessed, as well as when the coaching intervention takes place, and a statement in the methods clearly defining these terms. A timeline figure (or adapting figure 1) may also be useful if the scheme is complex.

- The authors refer to a single data coaching visit, but multiple follow-up visits. Presumably these additional visits also comprise feedback being given, so should these be considered as coaching visits as well?

- The structure of the manuscript could be improved. While I appreciate that this paper is presented as a case study, the attempt to conduct an evalaution of the coaching approach rather than simply report what was done means there is a need to clearly articulate some key details that would be part of a normal methods section. Perhaps a methods section with sub-titles including: study site / facility selection, intervention implementation/description/timeline, evaluation design, outcome indicators, analysis approach would help to improve flow.

Specific comments

- Line 90. I’m not familiar with “regional holistic assessments”. If this mechanism is particularly important or innovative, it could be helpful to add either a brief description or reference.

- Line 98. Some additional information in the background about the extent to which planning and implementation of malaria control policies are decentralized in Ghana would be helpful for context. The following paragraph (lines 99-117) focusses on data being reported up, but doesn’t elaborate on what kind of responses or decisions are made at the level of the CHPS.

- Line 102. Are “monitoring charts to guide clinical meetings” individual patient charts, or aggregate case count numbers?

- Line 144. Facilities needed to meet both the error rate >30% AND the data completeness rate <90%? Should this be “or”?

- Line 146. What kind of mistake generates an error flag? Did the validation exercise involve cross-referencing reported totals against OPD registers (presumably not since 605 facilities are mentioned later as the denominator)? It would also be helpful to know roughly how many indicators were validated – is this 3, 10, 100?

- Line 149. It is not clear how these three indicators relate to either the error rate of completeness rate. Were error rate and completeness only assessed for these indicators?

- Lines 160. Should this be “all prioritized facilities”?

- Line 203. “National DHIMS2 data” suggests that you used national aggregates of your indicators to evaluate the coaching program, which presumably was not the approach.

- Lines 205-208. It is not clear how monthly data were handled to compare between the periods. Did you simply aggregate, or take the value for the month immediately before the next visit, or generate monthly means for each phase?

- Lines 212-213. This statement implies that the facilities had to perform some additional data entry procedures for the evaluation on top of their usual responsibilities, which would bias any assessment of changes in data quality/completeness. Did the evaluation team actually download data from the DHIMS2 system?

- Line 225. I do not recall seeing specific dates when other phases were completed. How does this align with the overall implementation schedule?

- Line 237. There has not been any previous explanation of how overall facility scores are generated. Are different components weighted? Is it a simple arithmetic sum of all questions with a “correct” response?

- Line 242. What are the case management quality indicators? To my knowledge, there are not any indicators in DHIMS2 that directly assess quality of case management since this can only be assessed by independent observation of consultations.

- Lines 243. Please provide further details on how a paired t-test is conducted with data from 4 different periods. Are all comparisons made against the baseline or each each phase compared with the phase immediately prior?

- Line 255. Is the facility-level DHIMS2 data truly publicly available? If so, this is very unusual but a fantastic resource for other researchers, and warrants inclusion of a link in the manuscript. Or do you mean it is available to authorized users only?

- Line 256-257. It is unclear if these demographic details or coaching scores are included in the evaluation, and if so, how individual-level data have been summarized to facility-level.

- Line 275-276. Was the change in individuals a result of staff turnover, or intentionally targeting different team members on each visit?

- Table 1. The interpretation of numbers in brackets is not explained in the column titles.

- Table 2. The contents of these five dimensions have not been presented elsewhere, which limits the ability to interpret (or assess the validity of) this table.

- Table 3. Are the error rates the number of indicators that were not matching, or is this the empirical difference between a selected indicator in OPD registers and DHIMS2?

- Figure 2 is low quality and looks to be a screenshot or similar. It may be worthwhile re-generating this figure in an alternative software.

- The supplementary figure resolution means that the small font text is unreadable. While it’s nice to see an example that’s been completed, you may consider also including a copy of the template where all the text is readable.

6. PLOS authors have the option to publish the peer review history of their article (what does this mean? ). If published, this will include your full peer review and any attached files.

**Do you want your identity to be public for this peer review?** For information about this choice, including consent withdrawal, please see our Privacy Policy .

Reviewer #1: No

---

## [Decision Letter · Decision Letter 1]

30 Apr 2025

Coaching visits and supportive supervision for primary care facilities to improve malaria service data quality in Ghana: an intervention case study

PGPH-D-24-01778R1

Dear Dr Haws,

We are pleased to inform you that your manuscript 'Coaching visits and supportive supervision for primary care facilities to improve malaria service data quality in Ghana: an intervention case study' has been provisionally accepted for publication in PLOS Global Public Health.

Best regards,

Henry Surendra, PhD

Academic Editor

Reviewer Comments (if any, and for reference):

Reviewer's Responses to Questions

**Comments to the Author**

1. If the authors have adequately addressed your comments raised in a previous round of review and you feel that this manuscript is now acceptable for publication, you may indicate that here to bypass the “Comments to the Author” section, enter your conflict of interest statement in the “Confidential to Editor” section, and submit your "Accept" recommendation.

Reviewer #1: All comments have been addressed

2. Does this manuscript meet PLOS Global Public Health’s publication criteria ? Is the manuscript technically sound, and do the data support the conclusions? The manuscript must describe methodologically and ethically rigorous research with conclusions that are appropriately drawn based on the data presented.

Reviewer #1: Yes

3. Has the statistical analysis been performed appropriately and rigorously?

Reviewer #1: Yes

4. Have the authors made all data underlying the findings in their manuscript fully available (please refer to the Data Availability Statement at the start of the manuscript PDF file)?

Reviewer #1: Yes

5. Is the manuscript presented in an intelligible fashion and written in standard English?

Reviewer #1: Yes

6. Review Comments to the Author

Reviewer #1: (No Response)

7. PLOS authors have the option to publish the peer review history of their article (what does this mean? ). If published, this will include your full peer review and any attached files.

**Do you want your identity to be public for this peer review?** For information about this choice, including consent withdrawal, please see our Privacy Policy .

Reviewer #1: No
